# Disease burden and clinical severity of the first pandemic wave of COVID-19 in Wuhan, China

Juan Yang[1,6], Xinhua Chen[1,6], Xiaowei Deng[1], Zhiyuan Chen[1], Hui Gong[1], Han Yan[1], Qianhui Wu[1], Huilin Shi[1], Shengjie Lai [1,2], Marco Ajelli [3,4], Cecile Viboud[5] & Prof Hongjie Yu [1✉]

The novel coronavirus disease 2019 (COVID-19) was first reported in Wuhan, China, where the initial wave of intense community transmissions was cut short by interventions. Using multiple data sources, here we estimate the disease burden and clinical severity by age of COVID-19 in Wuhan from December 1, 2019 to March 31, 2020. Our estimates account for the sensitivity of the laboratory assays, prospective community screenings, and healthcare seeking behaviors. Rates of symptomatic cases, medical consultations, hospitalizations and deaths were estimated at 796 (95% CI: 703–977), 489 (472–509), 370 (358–384), and 36.2 (35.0–37.3) per 100,000 persons, respectively. The COVID-19 outbreak in Wuhan had a higher burden than the 2009 influenza pandemic or seasonal influenza in terms of hospitalization and mortality rates, and clinical severity was similar to that of the 1918 influenza pandemic. Our comparison puts the COVID-19 pandemic into context and could be helpful to guide intervention strategies and preparedness for the potential resurgence of COVID-19.

[1] School of Public Health, Key Laboratory of Public Health Safety, Ministry of Education, Fudan University, 200030 Shanghai, China. [2] WorldPop, Department of Geography and Environment, University of Southampton, University Road, Southampton SO17 1BJ, UK. [3] Department of Epidemiology and Biostatistics, Indiana University School of Public Health, Bloomington, IN 47405, USA. [4] Laboratory for the Modeling of Biological and Socio-technical Systems, Northeastern University, Boston, MA 02115, USA. [5] Division of International Epidemiology and Population Studies, Fogarty International Center, National Institutes of Health, Bethesda, MD 20892, USA. [6] These authors contributed equally: Juan Yang, Xinhua Chen. ✉email: yhj@fudan.edu.cn

As of 26 July 2020, 188 countries have been affected by the novel coronavirus disease 2019 (COVID-19), with 15,745,102 COVID-19 cases and 644,661 deaths reported worldwide[1]. COVID-19 has a broad spectrum of severity. Of serological-confirmed infections, only a fraction will develop symptoms. A fraction of symptomatic cases may seek medical care, when they can be identified via surveillance systems, require hospitalization, and die. Hospitalization is an important metric as it determines the strain exerted by an epidemic on the health care system. Further, deaths are highly relevant to planning pandemic responses, as mortality is an outcome that health authorities typically aim to minimize (Fig. 1a).

Estimates of disease burden and clinical severity of COVID-19 are critical to identify appropriate intervention strategies, plan for healthcare needs, and ensure the sustainability of the health system throughout the duration of the pandemic. However, quantifying these estimates based on surveillance data is challenging due to changes in health seeking behaviors during the pandemic, as well as underdiagnoses of a novel pathogen.

Historically, two influenza pandemics had far-reaching influences to humankind worldwide: the 1918 and 2009 influenza pandemics[2]. The 1918 influenza pandemic is typically considered as the worst-case pandemic scenario for pandemic planning. In contrast, the 2009 influenza pandemic is considered mild but provides a benchmark for a pandemic in modern times, as the health systems, supportive care, and populations, are comparable with those of today. Comparing the COVID-19 burden and clinical severity with past influenza pandemics can help public health officials interpret the magnitude of the COVID-19 pandemic and the success of the response efforts. A further comparison between the COVID-19 pandemic and seasonal influenza can be useful to optimize health resource allocations, considering their overlapping circulation periods.

Wuhan is a particularly well-suited location to assess the health burden of COVID-19. Firstly, Wuhan experienced intense community transmissions of severe acute respiratory syndrome coronavirus 2 (SARS-CoV-2); secondly, the first wave has ended, with only seven sporadic cases reported between March 24 and May 18[3]. Therefore, the first epidemic wave in Wuhan (for the period December 1, 2019–March 31, 2020) is an opportunity to comprehensively quantify the disease burden and clinical severity of COVID-19. Here, we used multiple data sources to estimate age-specific rates of symptomatic SARS-CoV-2 infections, medically attended cases, hospitalizations, and deaths, accounting for health seeking behaviors and underdiagnoses. We also estimated rates of medically attended influenza-like-illness (ILI) associated with SARS-CoV-2 infections; hospitalizations with severe acute respiratory infections (SARI), and pneumonia hospitalizations associated with SARS-CoV-2 infections by dividing the number of ILI consultations, SARI hospitalizations and pneumonia hospitalizations by the number of symptomatic SARS-CoV-2 infections. Moreover, we estimated the clinical severity of COVID-19 including the symptomatic case-fatality risk (sCFR), medically attended case-fatality risk (mCFR), hospitalization-fatality risk (HFR), symptomatic case-hospitalization risk (sCHR), and medically attended case-hospitalization risk (mCHR). The rates of symptomatic cases, medically attended cases, hospitalizations, and deaths with SARS-CoV-2 were calculated by dividing the number of cases at each level of severity by population size. Clinical severity was obtained by dividing the numbers of cases in the corresponding severity pyramid (Fig. 1a). Finally, we compared our estimates with those of the 1918 and 2009 influenza pandemics, and with seasonal influenza.

## Results

**Reported COVID-19 cases**. We obtained the number of laboratory-confirmed COVID-19 cases and clinically-diagnosed cases in Wuhan from published literatures and the Hubei Health Commission[3,4]. Cases were mainly confirmed by real-time reverse transcription polymerase chain reaction (RT-PCR) and included mild, moderate, severe, and critical cases[5–8]. Mild cases refer to cases with mild symptoms and no radiographic evidence of pneumonia. Moderate cases refer to cases with fever, respiratory symptoms, and radiographic evidence of pneumonia. Severe cases refer to cases with any breathing problems, finger oxygen saturation, and low $PaO_2/FiO_2$ ($PaO_2$ denotes partial pressure of oxygen in arterial blood; $FiO_2$ denotes fraction of inspired oxygen), etc. Critical cases refer to cases having any respiratory failures, shock, and any other organ failures that requires ICU admission. Clinically-diagnosed cases included suspected cases with pneumonia as indicated by chest radiography, but without virological confirmation of infection[6]. (Supplementary Information File 1) These clinically-diagnosed cases were included in our study, recognizing the value of a clinical definition at the peak of a pandemic and in the context of limited laboratory testing capacity. A total of 50,333 COVID-19 cases were reported in the 4-month epidemic in Wuhan. Of them, 32,968 (65.5%) were laboratory-confirmed cases. As of July 20, 3869 cases have died, and all others recovered. These cases were recorded from passive surveillance which was launched at the start of the outbreak in late December 2019 in Wuhan[9], and from active door-to-door and individual-to-individual screenings for fever (Supplementary Information File 2)[10,11].

**Estimated disease burden of COVID-19**. RT-PCR sensitivity for SARS-CoV-2 detection varied based on the interval between symptom onset and laboratory testing, which was highest (97.9%) at an interval of <7 days[12]. A population-based telephone and online survey conducted in Wuhan found that 35.4% (95% CI 28.4–43.9%) of patients with acute respiratory infections (ARI) (i.e., fever with any symptoms of cough, and/or sore throat) sought medical care during the epidemic of COVID-19[13]. All cases from passive surveillance were considered as medically attended cases. In the baseline analysis, we assumed that a proportion of mild cases, and all moderate-to-critical cases (had radiographic evidence of pneumonia) captured by active screenings in the community would eventually seek medical care given that the health system was not overwhelmed. It was assumed that the cases from passive surveillance had the same health seeking behavior as those captured by active screenings in the community. Laboratory-confirmed cases (moderate-to-critical) and clinically-diagnosed cases had radiographic evidence of pneumonia, and thus were considered as requiring hospitalizations (Fig. 1b).

After adjusting for sensitivity of RT-PCR testing, and accounting for the probability of seeking medical care and prospective screenings in the community, we estimated that a total of 52,300 (95% CI 50,500–54,500) medically-attended cases and 39,600 (95% CI 38,300–41,100) hospitalizations were associated with COVID-19. In Wuhan, over the period from December 2019 to March 2020, the rates of symptomatic cases, medical consultations, hospitalizations and deaths for COVID-19 were 796 (95% CI 703–977), 489 (95% CI 472–509), 370 (95% CI 358–384) and 36.2 (95% CI 35.0–37.3) per 100,000 individuals respectively. A consistent increasing trend with age was observed across all metrics, with the highest rates occurring in adults aged 60 years and over (Figs. 2a, 3a, 4a and Supplementary Information File 3).

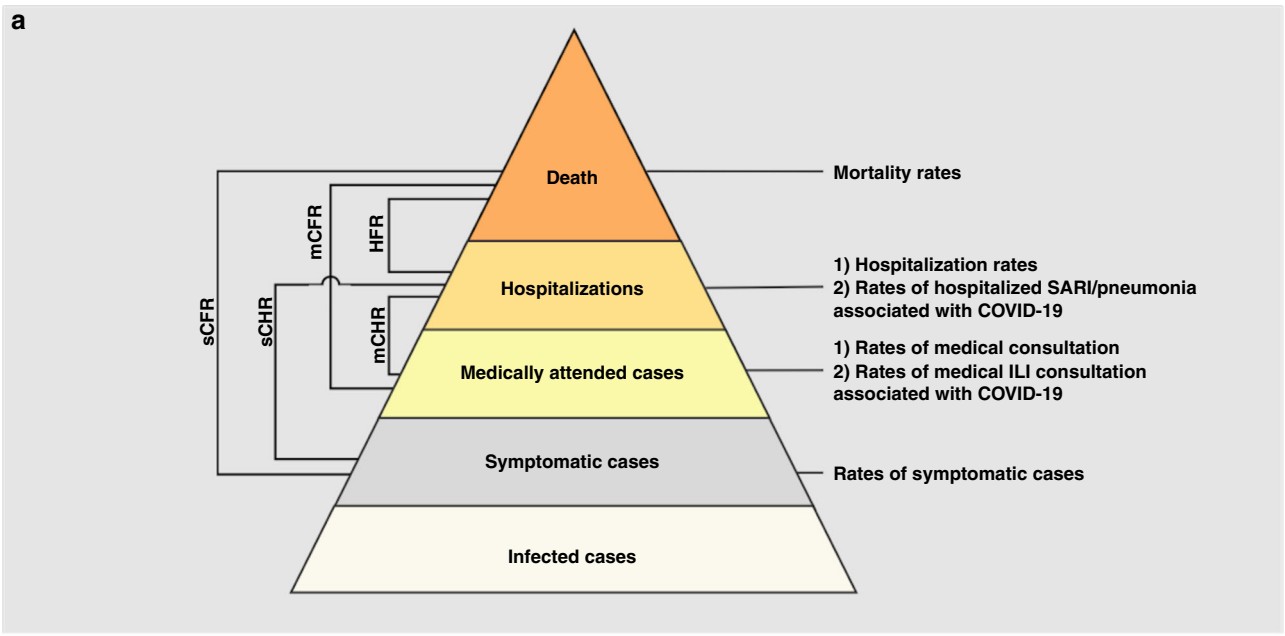

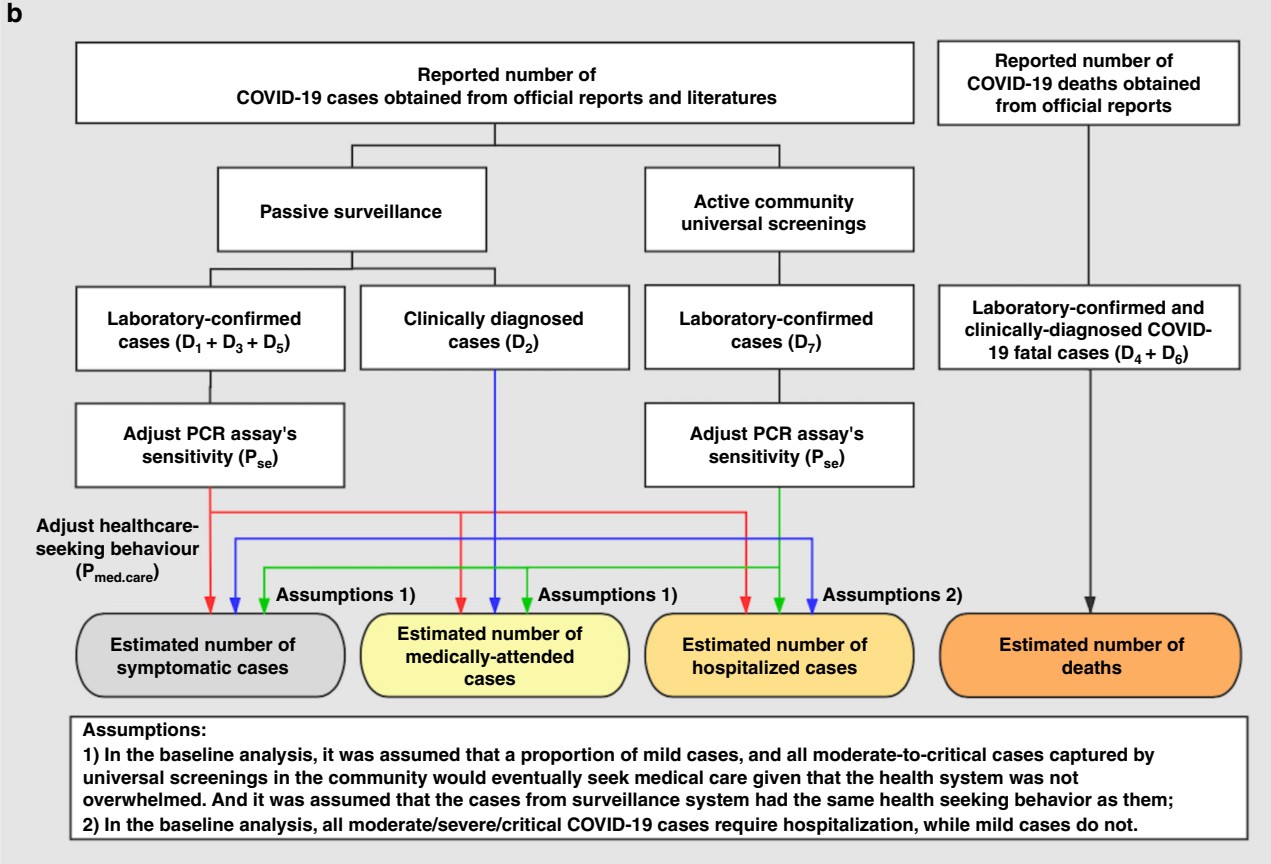

**Fig. 1 Severity levels of COVID-19 and schematic diagram of the baseline analyses. a** Severity levels of infections with SARS-CoV-2 and parameters of interest. Each level is assumed to be a subset of the level below. sCFR symptomatic case-fatality risk, sCHR symptomatic case-hospitalization risk, mCFR medically attended case-fatality risk, mCHR medically attended case-hospitalization risk, HFR hospitalization-fatality risk. **b** Schematic diagram of the baseline analyses. Data source of COVID-19 cases in Wuhan: D1) 32,583 laboratory-confirmed COVID-19 cases as of March 8[4], D2) 17,365 clinically-diagnosed COVID-19 cases during February 9–19[4], D3)daily number of laboratory-confirmed cases on March 9–April 24[3], D4) total number of COVID-19 deaths as of April 24 obtained from the Hubei Health Commission[3], D5) 325 laboratory-confirmed cases and D6) 1290 deaths were added as of April 16 through a comprehensive and systematic verification by Wuhan Authorities[3], and D7) 16,781 laboratory-confirmed cases identified through universal screening[10,11]. $P_{se}$: RT-PCR sensitivity[12]. $P_{med.care}$: proportion of seeking medical assistance among patients suffering from acute respiratory infections[13]. (Red, blue, and green arrows separately denote the data flow from laboratory-confirmed cases of passive surveillance, clinically-diagnosed cases, and laboratory-confirmed cases of active screenings).

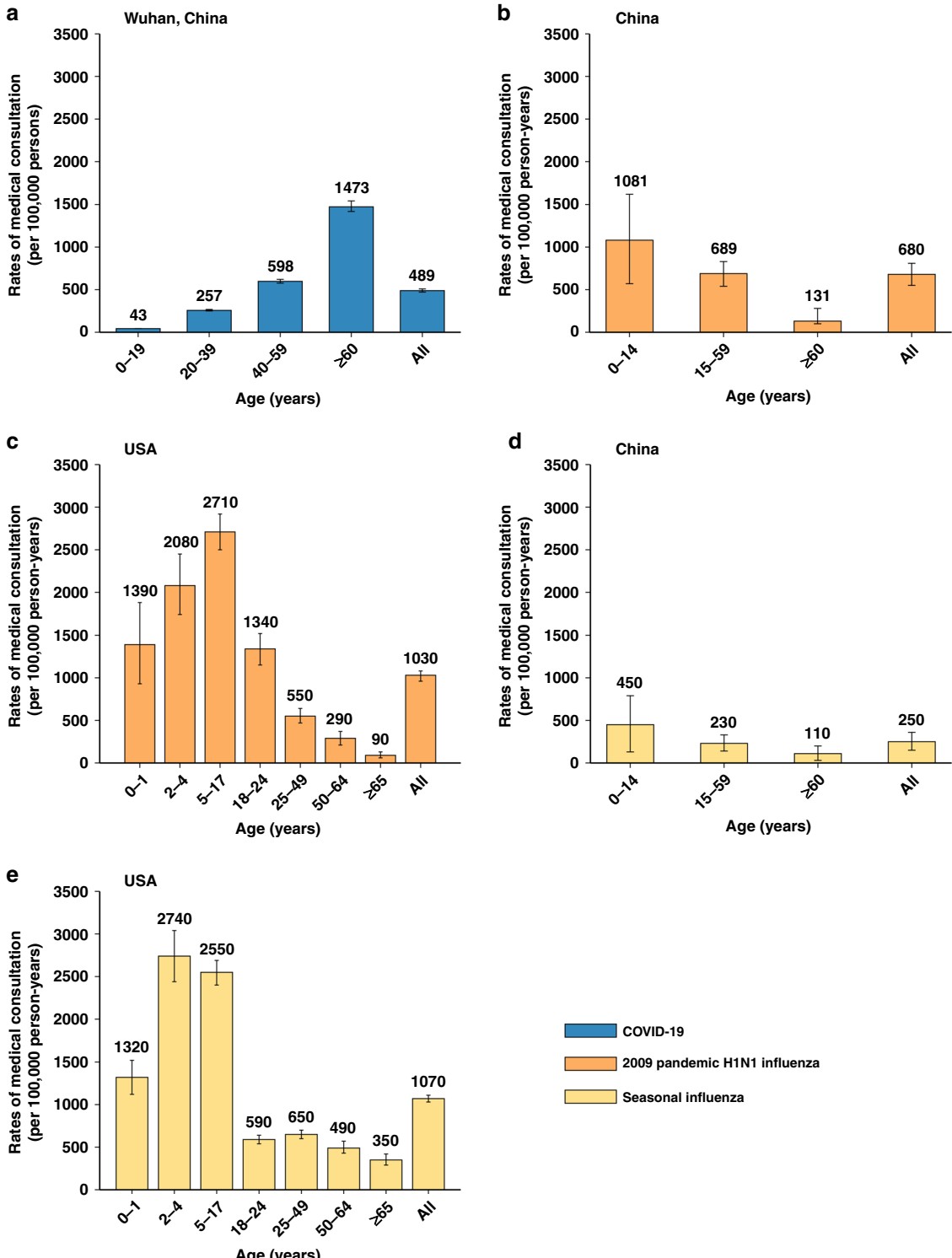

**Fig. 2 Rates of symptomatic cases and of medical consultation rates by age group (mean, 95% CI). a** Rates of medical consultation associated with COVID-19 in Wuhan, China. The error bars presented 95% CI as estimated using Monte Carlo sampling (10,000 samples from Binomial distributions). **b** Rates of medical consultation associated with 2009 pandemic H1N1 influenza, China[14]. **c** Rates of medical consultation associated with 2009 pandemic H1N1 influenza, USA[15]. **d** Seasonal influenza-associated excess ILI outpatient consultations rates, China[14]. **e** Rates of medical consultation associated with seasonal influenza, USA[15].

The rate of medical consultations for COVID-19 (mean: 489 per 100,000 individuals) was lower than that of the 2009 influenza pandemic in China and the US (680 and 1030 per 100,000 individuals, respectively). The rate of medical consultations was intermediate between that of the 2012–2013 influenza season in the US (1070 per 100,000 persons) and the 2006–2015 influenza seasons in China (mean: 250 per 100,000 individuals per year)[14,15]. (Fig. 2a–e and Supplementary Information Files 3, 4)

The hospitalization rates of COVID-19 in Wuhan were 3.1-fold higher than that of the 2009 influenza pandemic, and 1.8–2.6

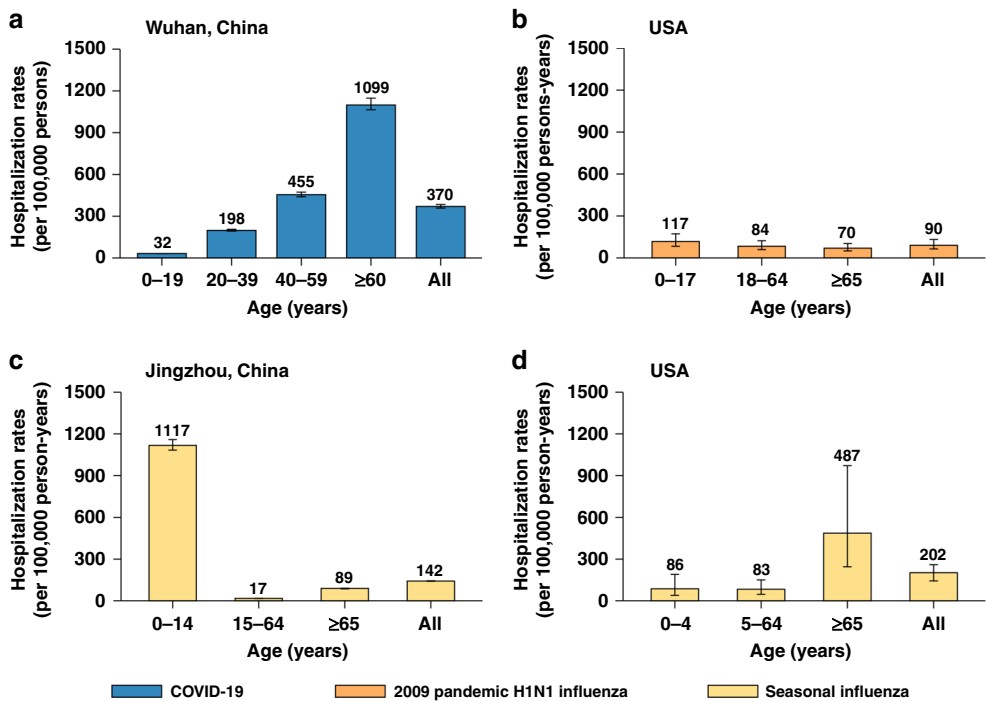

**Fig. 3 Hospitalization rates. a** Rates of hospitalization associated with COVID-19 in Wuhan, China (mean, 95% CI). The error bars presented 95% CI as estimated using Monte Carlo sampling (10,000 samples from Binomial distributions). **b** Rates of hospitalization associated with 2009 pandemic H1N1 influenza, USA (median, range)[16] **c** Rates of hospitalization associated with seasonal influenza related SARI in Jingzhou, Hubei province, China (median, range)[17]. **d** Rates of hospitalization associated with seasonal influenza, USA (mean, 95% CI)[18,19].

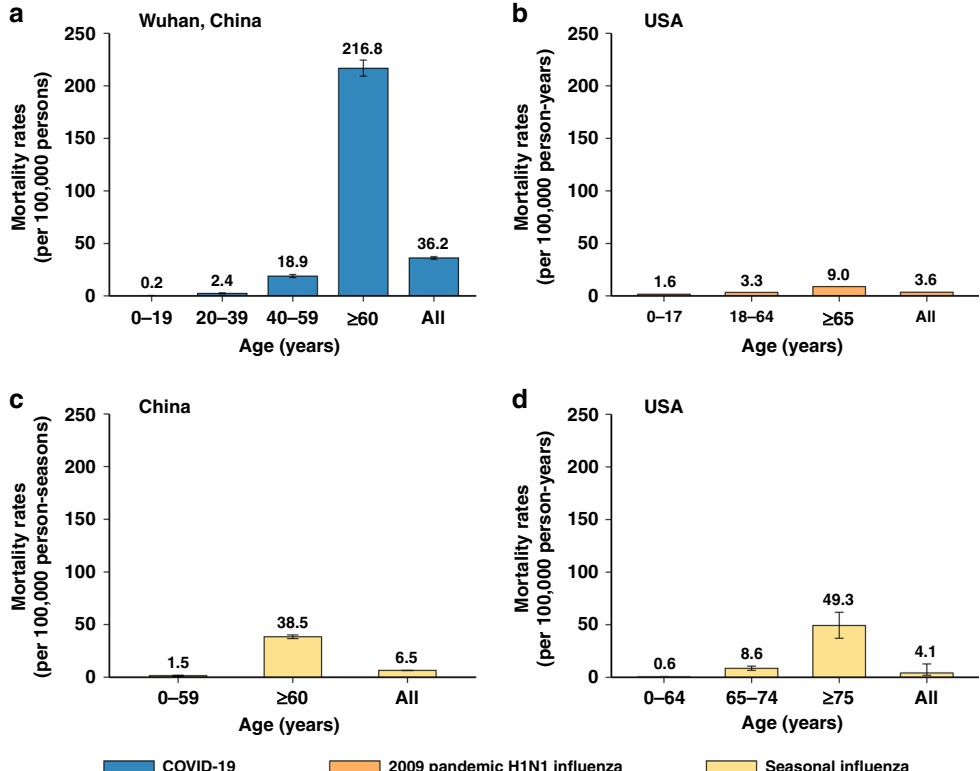

**Fig. 4 Mortality rates. a** Rates of mortality associated with COVID-19 in Wuhan, China (mean, 95% CI). The error bars presented 95% CI as estimated using Monte Carlo sampling (10,000 samples from Binomial distributions). **b** Rates of mortality associated with 2009 pandemic H1N1 influenza, USA (75% percentile)[20]. **c** Excess mortality rates associated with seasonal influenza, China (mean, 95% CI)[21]. **d** Excess mortality rates associated with seasonal influenza, USA (median, 95% credibility interval)[22].

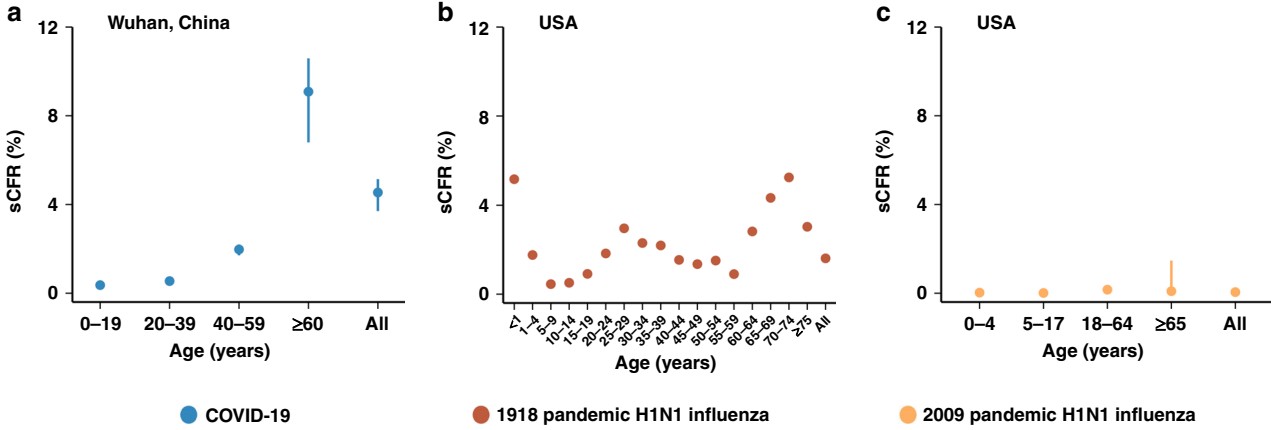

**Fig. 5 Symptomatic case-fatality risk (sCFR). a** sCFR associated with COVID-19 in Wuhan, China (mean, 95% CI). The error bars presented 95% CI as estimated using Monte Carlo sampling (10,000 samples from Binomial distributions). **b** sCFR associated with 1918 pandemic H1N1 influenza in August–December 1918, USA (mean)[23]. **c** sCFR associated with 2009 pandemic H1N1 influenza, USA (median, 95% CI)[25].

times that of seasonal influenza[16–19]. Higher hospitalization burden was found among older adults for COVID-19, while the hospital burden was shifted towards children for seasonal influenza in China[17] and the 2009 influenza pandemic in the US[16]. (Fig. 3 and Supplementary Information Files 3, 4) The overall mortality rate of COVID-19 in Wuhan was much higher than that of 2009 influenza pandemic and seasonal influenza (36.2 vs. 3.6–6.5 per 100,000 individuals)[20–22]. (Fig. 4 and Supplementary Information Files 3, 4)

**Estimated clinical severity of COVID-19.** The overall sCFR of COVID-19 was 4.54% (95% CI 3.70–5.14%), which is comparable, if not higher, to that of the 1918 influenza pandemic—from the analysis of data from eight US localities, the sCFR was estimated at 1.61% and 1.98% for the first and second wave, respectively[23,24]. Such a figure is substantially higher than that of the 2009 influenza pandemic (<0.1% in the US)[25]. The sCFR of COVID-19 was higher for adults aged ≥60 years than for the other age groups (9.09% vs. 0.36–1.97%). (Fig. 5a–c and Supplementary Information File 5) In contrast, younger age groups were the most affected segment of the population during the 1918 and 2009 influenza pandemic, while both young and old individuals were the most affected during seasonal influenza epidemics[14–26].

The HFR (9.77%, 95% CI 9.41–10.10%) and sCHR (46.48%, 95% CI: 39.33–50.93%) were higher for COVID-19 than for the 2009 influenza pandemic (HFR: 2.6% in North America[26]; sCHR: 1.44% in the USA[25]) (Supplementary Information Files 3, 5).

**Sensitivity analyses.** To assess the robustness of our findings, we conducted four sensitivity analyses: In scenario (i) we assumed that moderate cases had the same health seeking behavior as mild cases, i.e., only a proportion of moderate cases sought medical assistance, in scenario (ii) we excluded clinically-diagnosed cases, in scenario (iii) we used the upper limit of 95% CI of the probability of seeking medical care, and in scenario (iv) we used the lower limit of 95% CI of the probability of seeking medical care. Compared to the baseline analysis, the mean rates of symptomatic cases for COVID-19 increased from 796 to 935 per 100,000 persons in scenario (i), and 960 per 100,000 persons in scenario (iv), while rates decreased to 634 per 100,000 persons in scenario (ii), and 719 per 100,000 persons in scenario (iii). The sCFR decreased from 4.54 to 3.87% in scenario (i) and 3.77% in scenario (iv), while it increased to 5.38% in scenario (ii) and 5.03% in scenario (iii). Similar patterns were observed for the other metrics

of interests (Supplementary Information File 3). Overall, the estimated variations did not change our findings, particularly for comparison of COVID-19 with influenza pandemics and seasonal influenza.

**ILI consultations and SARI/pneumonia hospitalizations.** We quantitatively assessed the impact of COVID-19 on the healthcare sector using the local number of ILI consultations, and SARI/pneumonia hospitalizations in the absence of COVID-19 as a reference. The rate of medically attended COVID-19 was approximately 1.3 times the baseline ILI consultations among adults of ≥60 years of age. The hospitalization rate of COVID-19 was 3–6-fold higher than baseline SARI hospitalizations among adults of ≥20 years of age, and 25–132 fold higher than pneumonia hospitalizations as a reference (Supplementary Information File 6).

**Discussion**

This study used multiple sources of data to estimate different levels of the COVID-19 severity pyramid. We found that the mean rates of symptomatic cases, medical consultations, hospitalizations and deaths were respectively 796, 489, 370, and 36.2 per 100,000 persons in Wuhan from December 2019 to March 2020. All burden metrics increased with age, with adults ≥60 years of age most affected. Similarly, the highest sCFR and HFR were found in older adults.

Our study was strengthened by adjustment for several potential biases. First, rates of medical consultations were adjusted by the sensitivity of RT-PCR assays[12]. Sensitivity was only 30–40% before January 23 due to delayed detection, which could lead to important underdiagnoses and has not been considered in previous studies. Second, we accounted for the health seeking behaviors among the Wuhan population during the epidemic[13]. The probability of seeking medical treatment conditionally on symptoms of acute respiratory diseases is a critical parameter to estimate the true number of COVID-19 cases in community. Accordingly, our estimates of disease burden may be the most accurate for Wuhan so far.

Two studies reported on COVID-19 disease burden in the US and Canada by the end of May, at which time local epidemics are still ongoing (Supplementary Information File 7)[27,28]. The overall rate of symptomatic cases (796 vs. 404–534 per 100,000 persons) in Wuhan was much higher than that in the US and Québec, a severely affected province of Canada. Variation in testing strategies likely contribute to the difference in rate of symptomatic

cases, in addition to true difference in epidemic dynamics. Unlike in Wuhan, only individuals with signs or symptoms consistent with COVID-19, and asymptomatic individuals with suspected exposure were preferentially tested in the US. Moreover, in contrast to our study, the US and Canadian estimates were not adjusted for the sensitivity of RT-PCR assays and health seeking behaviors, and thus may be underestimated.

Our estimated hospitalization rate for a 4-month COVID-19 outbreak was much higher than that for a 3–6-month COVID-19 outbreak in the US and Québec (370 vs. 50–114 per 100,000 persons)[27–30]. We estimated that 76% of medically-attended cases were hospitalized in Wuhan, while only 18% were hospitalized in the US[30]. The difference between these estimates could be explained by the potential different clinical thresholds for hospitalization. We assumed that moderately ill cases with radiographic evidence of pneumonia and more severe cases would be hospitalized in the context of medical practices in China, based on the probability of progression from disease to death[31]. However, such results may not apply to other countries with different healthcare practice and general health seeking behaviors. For example, Chinese patients are less likely to seek care at primary health institutions than hospitals[32].

The mortality rate of COVID-19 in Wuhan was lower than the excess mortality in New York City (24,172 excess deaths[33] among 19,746,286 individuals, and thus 122 per 100,000 individuals), and Québec (52 per 100,000 persons)[28], but similar to the national excess all-cause mortality in the US (122,300 excess deaths among 332,382,720 individuals, and thus 36.8 per 100,000 individuals)[34]. In our study, patients who died at home or died before being diagnosed were not considered, which could have been important in the early phase of the epidemic due to under-ascertainment. A recent comprehensive correction of official tallies of cases and deaths by the Wuhan Authorities, included in this study, could minimize this under-ascertainment. Further research should use excess mortality approaches[33] to capture the full burden of the outbreak, when vital registration data for this period become available.

Our estimates of sCFR (4.54% vs. 1.2–1.4%) and mCFR (7.40% vs. 5.91%) for Wuhan were higher than in prior modeling studies[35,36]. This was likely explained by the addition of revised statistics on cases and deaths, and a more complete dataset with no right-censored outcomes in our study. Large variations in mCFR were observed between countries, which have not been systematically analyzed. Qualitatively, these variations could be explained by differences in the sensitivity of surveillance systems to detect cases at different levels of the severity pyramid, differences in clinical care of severe and critical patients, and in age structure and underlying conditions of the population.

Our HFR estimate (9.77%) was higher than the estimate obtained by Wang et al. in a highly censored sample in Wuhan in the very early stages of the epidemic (4.3%)[37]. However, it was much lower than the 28% estimate obtained in two COVID-19-designated hospitals for severe COVID-19 cases in Wuhan, probably due to the particularly high proportion of severe and critical patients hospitalized in these facilities (64% vs. 28%)[38]. Our HFR estimate was lower than the 18.1% estimate in France[39], probably due to aforementioned loose threshold for hospital admissions in China and preference of seeking care in hospitals rather than outpatient settings.

We systematically compared the burden of the COVID-19 outbreak with that of the 1918 and 2009 influenza pandemics and seasonal influenza. Our COVID-19 estimates were substantially higher than those of the 2009 influenza pandemic and seasonal influenza, and similar to the 1918 influenza pandemic. However, the age pattern of severe disease was clearly different. Our COVID-19 severity estimates increased with age. In contrast,

younger age groups were the most affected by 1918 and 2009 influenza pandemic, and both youngest and oldest individuals were the most affected by seasonal influenza. Small changes were observed when we adjusted the overall burden and clinical severity of seasonal and pandemic influenza using Wuhan age profile as a reference (Supplementary Information Files 5 and 8). Comparison of severity estimates between pandemics was difficult to standardize, particularly for 1918 influenza pandemic[23]. The 1918 sCFR was based on data from a single US study from more than a 100 years ago, at a time when awareness of viral diseases was inexistent, case ascertainment and disease surveillance were limited, and definition of clinical outcomes varied. Therefore, our comparison was not intended to quantify the absolute value of differences, but to put the COVID-19 pandemic into perspective.

To put our results in perspective, it is important to stress that our COVID-19 estimates refer to the first epidemic wave in Wuhan—a 4-month long period. The epidemic was controlled by intense interventions[4]. If the epidemic rebounds, as one would expect if the infection was reintroduced in a population with low immunity, the disease burden would rise. Moreover, given that the epidemic lasted only 4 months, the stress on the healthcare system was tremendous, as severe cases and hospitalizations were concentrated over a relatively short period of time. Furthermore, neither seasonal nor pandemic influenza outbreaks were controlled, as vaccination was either low or delayed until after the main wave had passed, and no social distancing was put in place.

Using a simple data-driven approach, we quantitatively assessed the impact of COVID-19 on the healthcare sector using the local number of ILI consultations, and SARI/pneumonia hospitalizations in the absence of COVID-19 as a reference. The number of COVID-19 hospitalizations was several folds higher than that of baseline SARI hospitalizations and 25–132 folds higher than that of pneumonia hospitalizations among adults (≥20 years of age). This indicates that during this time period, the Wuhan healthcare system considerably exceeded surge capacity, highlighting the importance and necessity of preparedness for sufficient healthcare resources. Moreover, there is a winter peak of consultations and hospitalizations related to respiratory diseases, such as seasonal influenza and respiratory syncytial virus[14,40,41], which may have contributed to overwhelm the healthcare sector during the first wave of the COVID-19 epidemic.

Our study has some limitations. First, health-seeking behavior may have not remained constant throughout the epidemic. In this survey, study participants in Wuhan were asked to review their history of ARI between December 2019 and March 2020, and whether they sought medical assistance for these symptoms[13]. However, since we did not obtain the onset date of these symptoms, and hence we could not stratify health-seeking behavior by COVID-19 epidemic phase. Instead, we calculated the overall proportion of ARIs cases who sought medical care during the epidemic. If the distribution of onset dates of ARIs cases in our sample was skewed towards the early phase (late January) of the epidemic, the proportion seeking medical care may be underestimated due to the overwhelmed health system. That would lead to an overestimation of the number of symptomatic COVID-19 cases. Conversely, if the distribution of onset dates was skewed towards the late phases of epidemic, we may have underestimated the number of symptomatic COVID-19 cases. We conducted a sensitivity analysis on the probability of seeking medical care, using the lower and upper limits of the 95% CI of our survey. This analysis resulted in minor changes in the disease burden and clinical severity estimates compared to the baseline analysis.

Second, missing or incorrect records of COVID-19 cases are inevitable during an outbreak, particularly in the period when the healthcare capacity is overwhelmed. A verification of reported

COVID-19 cases was conducted by Wuhan Authorities to correct for late reporting, omissions, and misreporting, with 325 laboratory-confirmed cases and 1290 deaths added to official tallies. Due to the high specificity of RT-PCR assay (almost 100%), false positives were rare[42–45]. We included in the analysis clinically-diagnosed cases that were reported for a brief 1-week period, when testing could not keep up, so we inevitably over-estimated the true number of COVID-19 cases. Our sensitivity analyses showed that the exclusion of these clinically-diagnosed cases led to decreased estimates of disease burden and increased estimates of clinical severity. However, our conclusions were robust to these changes. Moreover, while reporting of cases changed at the beginning of the pandemic as the definition of COVID-19 suspected cases broadened to include a milder spectrum, we expected reporting was relatively stable throughout the rest of the outbreak although this would be difficult to prove conclusively.

Additionally, it is difficult to extrapolate our findings to other countries/regions since estimates of clinical severity and disease burden of COVID-19 are influenced by multiple factors such as the evolution of the epidemic, intervention policies, case detection strategy, surge capacity of healthcare systems, differences in presentation, triage, and treatment, and health seeking behaviors over time and across locations. A modelling study has revealed that containment has proved to be successful to control the local COVID-19 epidemic in Wuhan. Without containment efforts, the number of COVID-19 cases would have been an estimated 67-fold higher than that has been thus far[46]. Therefore, our estimates in Wuhan could represent the disease burden and clinical severity in a region with (1) wide-spread community transmission of SARS-CoV-2; (2) strict nonpharmaceutical interventions, referred as to "wartime measures" in the study by Leung et al.[47]; (3) extensive detection of all outpatients with fever[10]; (4) enhanced healthcare capacity. Indeed, Wuhan experienced remarkably rapid and extensive support from top-level medical staff drawn from all over China, as well as rapid establishment of medical facilities like the Leishenshan and Fangcang shelter hospitals[48–50]. Although the COVID-19 pandemic has already spread across the world, and the scale of epidemics in western countries, like the US and Brazil, exceeded that in Wuhan by far, the pandemic in other countries is still ongoing and any estimate is bound to be revised. Our estimates represent the full impact of a short but intense first wave, and could be considered as benchmarks to plan intervention strategies for a potential second wave of the pandemic.

In the first wave of the COVID-19 pandemic in Wuhan, China (December 2019–March 2020), intense community transmissions caused higher disease burden than the 2009 influenza pandemic and seasonal influenza. Overall, we find that the clinical severity of COVID-19 seems to be in the same order of magnitude as that of the 1918 influenza pandemic. In contrast to the age pattern of influenza virus infection, however, the highest burden and clinical severity of COVID-19 is observed among older adults, while children are less affected. During the epidemic of COVID-19, the Wuhan healthcare system considerably exceeded surge capacity. This study is helpful to guide intervention strategies and healthcare preparedness for the potential re-emergence of COVID-19 in China and beyond.

## Methods

**Case definitions**. Case definitions for laboratory-confirmed-cases were issued by the National Health Commission of China, and included mild, moderate, severe, and critical cases. Cases were confirmed by real-time reverse transcription polymerase chain reaction (RT-PCR) or by viral sequencing indicating genomes highly homologous to SARS-CoV-2[5–8]. Clinically-diagnosed cases included suspected cases with pneumonia as indicated by chest radiography, but without virological confirmation of infections[6]. The "clinical" definition was only used for 1 week in Hubei province as laboratory testing capacity was insufficient, and led to a large number of clinical cases to be isolated and treated without delay. These clinically-diagnosed cases were included in our study, recognizing the value of a clinical definition at the peak of a pandemic and in the context of limited laboratory testing capacity. The laboratory-confirmed cases include mild-to-critical cases, while the clinically-diagnosed cases include moderate-to-critical cases. Definitions are presented in detail in Supplementary Information File 1.

### Data source

*COVID-19 cases*. Our study aimed to account for underdiagnosis associated with the sensitivity of laboratory assays, which is strongly dependent on the time lag between symptom onset and diagnostic test[12]. The distribution of lags varied at different phases of the epidemic in Wuhan due to laboratory testing capacity[4]. Accordingly, the daily number of COVID-19 cases by symptom onset date was preferred to the aggregated cumulative data.

We obtained the following data: the daily number of laboratory-confirmed cases based on date of symptom onset in Wuhan extracted from a study which included 32,583 laboratory-confirmed COVID-19 cases as of March 8 (D1), and 17,365 clinically-diagnosed COVID-19 cases during February 9–19 (D2)[4]. The daily number of laboratory-confirmed cases in Wuhan based on reporting date on March 9–April 3 were extracted from the Hubei Health Commission (D3)[3]. The total number of COVID-19 deaths in Wuhan as of April 24 was obtained from the Hubei Health Commission (D4)[3]. To correct the late reporting, omissions and misreporting of COVID-19 cases due to the healthcare capacity being overwhelmed during the outbreak, Wuhan Authorities conducted a comprehensive and systematic verification of reported COVID-19 cases between late March and middle April. A total of 325 laboratory-confirmed cases (D5) and 1290 deaths (D6) were added on April 17[3]. The number of COVID-19 cases stratified by age and clinical category was obtained from the above sources data D1 and D2, while the age profile of fatal cases was obtained from the China CDC Weekly report[4,51].

All of these datasets were registered through a surveillance system, which was launched to record the information on COVID-19 cases in China at the start of the outbreak in late December 2019 in Wuhan[9]. These data were collected from passive surveillance, and active door-to-door and individual-to-individual screenings for fever. The active screening was implemented twice in Wuhan on a daily basis from January 24–February 10 to February 17–19[10,11]. A total of 16,781 laboratory-confirmed cases were identified through active screenings (D7, Details shown in Supplementary Information File 2)[10,11].

*Sensitivity of RT-PCR*. A study retrospectively analyzed the RT-PCR assays of 301 patients with 1113 specimens in Wuhan, and found that RT-PCR sensitivity varied at different phases of the epidemic due to the difference of interval between symptom onset and laboratory testing ($P_{se}$) (Supplementary Information File 9). The sensitivity of RT-PCR assays was highest (97.9%) at an interval of <7 days[12].

*Health seeking behavior surveys*. A population-based telephone and online survey was conducted to understand the health seeking behaviors of patients suffering from acute respiratory infections (i.e., fever with any symptoms of cough, and/or sore throat) during the epidemic of COVID-19 in Wuhan. Of patients with acute respiratory infections, 35.4% (95% CI 28.4–43.9%) sought medical care, by adjusting for the age structure of Wuhan population. Children had a higher probability of medical attendance than adults ($P_{med.care}$)[13,52].

*Other datasets*. A total of 10.7 million persons lived in Wuhan during the epidemic[53]. The age profile of the Wuhan population was obtained from the China Statistic Yearbook[54]. To compare the burden of COVID-19 to baseline activity of acute respiratory infections, we obtained reference historical data on ILI surveillance in Hubei province and SARI surveillance in Jingzhou city, Hubei province[14,17,21]. Additionally, we collected the annual number of consultations in pediatric and internal medicine departments in Hubei, and the national number of pneumonia hospitalization rates from the Chinese Health Statistics Yearbook[55]. All these data were collected from publicly available sources and did not contain any personal identifiable information. Summary of data were presented in Supplementary Information File 10.

**Statistical analysis**. Figure 1 described the metrics we estimated, data flow, data analysis procedure and assumptions in the baseline analyses. All analyses were performed in R version 3.6.3[56].

*Reported COVID-19 cases in Wuhan*. In the baseline analysis, we considered COVID-19 cases in Wuhan as those with laboratory-confirmation or with a clinical diagnosis (for the brief period where the clinical definition was in place) and tabulated data by symptom onset date. The interval between symptom onset and diagnosis was obtained from data D1[4]. Then, we randomly simulated 10,000 draws from a gamma distribution representing these time intervals to estimate onset dates for laboratory-confirmed cases reported between March 9 and April 3 (data D3), and added laboratory-confirmed cases (data D5). This allowed us to impute onset dates for cases that did not have this information.

*Medical consultations.* All cases from passive surveillance were considered as medical attendance (data D1 + D3 + D5, and D2). In the baseline analysis, we assumed that a proportion of mild cases, and all moderate-to-critical cases captured by active screenings in the community (data D7) would eventually seek medical care given that the health system was not overwhelmed (Assumption 1). The health seeking behavior of mild cases was assumed to be the same as aforementioned patients with acute respiratory infections during the COVID-19 epidemic ($P_{med.care}$)[13]. Hence, to estimate medically attended cases, we only excluded a proportion of $(1 − P_{med.care})$ mild cases identified by community screening from the total reported COVID-19 cases. Moreover, the number of laboratory-confirmed cases from official reports (data D1 + D3 + D5, and D7) was divided by the sensitivity of RT-PCR ($P_{se}$) to account for underdiagnoses.

*Symptomatic cases.* In the baseline analysis, we assumed the cases from surveillance system (data D1 + D3 + D5, and D2) had the same health seeking behavior as those captured by active screenings in the community (data D7) given that the health system was not overwhelmed (Assumption 1). Accordingly, the number of mild symptomatic cases was estimated by dividing reported mild COVID-19 cases by the probability of seeking medical care, conditionally on self-reported acute respiratory infection[13]. Adjustment of sensitivity of RT-PCR was considered as well.

*Hospitalized cases.* Moderate-to-critical COVID-19 cases had radiographic evidence of pneumonia, while mild cases were defined as those without radiographic evidence of pneumonia[5–8]. Chest x-ray confirmed pneumonia is a threshold for hospital admissions in China. Accordingly, in our study, estimates for SARS-CoV-2 related hospitalizations excluded patients defined as mild cases in the baseline analysis (Assumption 2).

In above analyses, to account for the uncertainty of two parameters (RT-PCR sensitivity and probability of seeking medical care), we conducted a Monte Carlo Simulation by drawing 10,000 samples from Binomial distributions. We generated 10,000 estimates for the number of COVID-19 cases, based on which we calculated the median, and 95% CIs (the 2.5th and 97.5th percentiles) for the outcomes of interest in this study.

Additionally, the following sensitivity analyses were conducted: in scenario (i) for above Assumptions 1 and 2, we assumed moderate cases had the same health seeking behavior as mild cases, i.e., only a proportion of moderate cases sought medical assistance ($P_{med.care}$); and in scenario (ii) we excluded clinically-diagnosed cases.

*Disease burden.* We used the number of ILI consultations, and the number of SARI/pneumonia hospitalizations in the absence of COVID-19 outbreak as a reference to estimate COVID-19 related ILI medical consultations, and COVID-19 associated SARI/pneumonia hospitalization rate. Estimation of the number of ILI cases and SARI/pneumonia hospitalizations during the periods were shown in Supplementary Information Files 11–14.

Moreover, for comparison with historical outbreaks, we conducted a narrative review on estimates of disease burden and clinical severity for the 1918 and 2009 influenza pandemics, as well as seasonal influenza in China and USA (Summary of studies shown in Supplementary Information Files 4, 5). The age profile of COVID-19 cases was obtained from data D1[4], in which COVID-19 cases were broken down into 20-year age categories. We could not generate disease burden and clinical severity estimates for influenza using the same age stratification, because numerators and denominators were not available from the literatures.

**Role of the funding source.** The funder of the study had no role in study design, data collection, data analysis, data interpretation, or writing of the report. The corresponding author had full access to all the data in the study and had final responsibility for the decision to submit for publication.

**Reporting summary**. Further information on research design is available in the Nature Research Reporting Summary linked to this article.

## Data availability

All aggregated data analyzed in this study are included in the Article and its Supplementary Table S8. Individual-level data on confirmed COVID-19 cases as of March 8, 2020 are available on GitHub at https://github.com/chaolongwang/SAPHIRE, and health seeking behavior survey data are available on GitHub at https://github.com/zychenfd/Wuhan-disease-burden[52]. All data supporting the findings of this study are available from the authors upon request.

## Code availability

The R code to replicate the analyses is available on GitHub at https://github.com/zychenfd/Wuhan-disease-burden[52].

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

## Acknowledgements

The study was supported by grants from the National Science Fund for Distinguished Young Scholars (No. 81525023), National Science and Technology Major Project of China (No. 2018ZX10201001-010, No. 2018ZX10713001-007, No. 2017ZX10103009-005). This study does not necessarily represent the views of the US government or the National Institutes of Health.

## Author contributions

H.Y. conceived, designed and supervised the study. J.Y., X.C., X.D., Z.C., H.G., Han.Y., Q.W., H.S., and S.L. participated in data collection. J.Y., X.C., X.D., Z.C., and H.G. analyzed the data, and prepared the tables and figures. J.Y. prepared the first draft of the manuscript. S.L., M.A., C.V., and H.Y. commented on the data and its interpretation, revised the content critically. All authors contributed to review and revision and approved the final manuscript as submitted and agree to be accountable for all aspects of the work.

## Competing interests

M.A. has received research funding from Seqirus and H.Y. has received research funding from Sanofi Pasteur, GlaxoSmithKline, Yichang HEC Changjiang Pharmaceutical Company, and Shanghai Roche Pharmaceutical Company. None of those research funding is related to COVID-19. All other authors report no competing interests.
