## [Peer Review File · Nature Communications]

Reviewers' Comments:

Reviewer #1:

Remarks to the Author:

The authors have satisfactorily responded to my concerns

John Drake

Reviewer #2:

Remarks to the Author:

This revised manuscript by Yang, et al., characterizing disease burden and severity of COVID-19 infection in Wuhan, China compared to influenza pandemics 1918 and 2009, is significantly improved. The authors appear to have adequately addressed both reviewers' concerns and have presented a revised manuscript which is more complete, clear, and better placed within the current context of the COVID-19 pandemic and updated literature. Additional clarifications and excess data have been moved to a fairly comprehensive supplement

Minor notes:

May be helpful to clarify the abstract when stating that COVID-19 had higher burden than 2009, perhaps can define burden earlier? Disease burden means cases, consultations, and hospitalizations (or just the first 2)?

Figure 5 may be a bit busy and difficult for readers to easily interpret. There is a mix of many different rates/outcomes, across age groups, comparing different locations and pandemics. The variable on the x-axes, despite all being age, use different age groups, the outcomes and scale vary for the y-axes, and some graphs report medians w/ CI while others report means. I understand the limitations of the data source and there is a lot of important information here. However, there is also so much variation among the different graphs that they essentially have to be digested almost one at a time. I wonder if finding a way to group outcomes or locations or pandemics together in a way that allows for easier comparisons might be possible. Or cutting down on the number of figures within Figure 5 to focus on the ones the authors find most salient

The authors seek to improve openness by providing additional data and code, but in multiple places it seems that placeholders still exist in the manuscript, where it reads "available on GitHub at XX". Please fill in the XX portion

Please perform a close read and double check for grammatical accuracy and consistency. For example, in the abstract, 'multiple data source' should be plural, and in the manuscript, there is some inconsistency in use of past and present tense.

As a minor suggestion, in working to address reviewer concerns, the manuscript has continued to grow in length. There may be opportunities to shorten or streamline parts of the paper, particularly the Introduction and Discussion, without sacrificing information and meaning.

Reviewers' comments:

Reviewer #1 (John Drake) (Remarks to the Author):

The authors have satisfactorily responded to my concerns.

Response: We would like to thank the reviewer for taking the time to review our manuscript once again. And we are grateful for his positive evaluation of our manuscript.

Reviewer #2 (Remarks to the Author):

1. This revised manuscript by Yang, et al., characterizing disease burden and severity of COVID-19 infection in Wuhan, China compared to influenza pandemics 1918 and 2009, is significantly improved. The authors appear to have adequately addressed both reviewers' concerns and have presented a revised manuscript which is more complete, clear, and better placed within the current context of the COVID-19 pandemic and updated literature. Additional clarifications and excess data have been moved to a fairly comprehensive supplement.

Response: We would like to thank the reviewer for taking the time to review our manuscript once again. We are grateful for her/his positive evaluation of our manuscript and the useful comments that helped us further improve our work.

Minor notes:

2. May be helpful to clarify the abstract when stating that COVID-19 had higher burden than 2009, perhaps can define burden earlier? Disease burden means cases, consultations, and hospitalizations (or just the first 2)?

Response: Thanks for the comment. Disease burden here means rates of hospitalizations and mortality. We revised in the abstract as suggested.

Page 3, line 38-40: The COVID-19 outbreak in Wuhan had higher burden than the 2009 influenza pandemic or seasonal influenza in terms of hospitalization and mortality rates.

3. Figure 5 may be a bit busy and difficult for readers to easily interpret. There is a mix of many different rates/outcomes, across age groups, comparing different locations and pandemics. The variable on the x-axes, despite all being age, use different age groups, the outcomes and scale vary for the y-axes, and some graphs report medians w/ CI while others report means. I understand the limitations of the data source and there is a lot of important information here. However, there is also so much variation among the different graphs that they essentially have to be digested almost one at a time. I wonder if finding a way to group outcomes or locations or pandemics together in a way that allows for

easier comparisons might be possible. Or cutting down on the number of figures within Figure 5 to focus on the ones the authors find most salient.

Response: The reviewer is correct that figure 5 looks busy and difficult for readers to easily interpret. As suggested, in the revised version of Figure 5 we focused on panels a-c, which present the comparisons of sCFR of COVID-19 pandemic, 1918 influenza pandemic, and the 2009 influenza pandemic. We deleted panel d (mCFR) and panel i (mCHR) and moved the information to the Supplementary Information File 3 (Table S2); further comparisons with influenza pandemics and seasonal influenza were not available. Moreover, we removed panels e and f (HFR of COVID-19 pandemic and 2009 influenza pandemic) and moved them to the Supplementary Information 5, Figures S1; we removed panels g and h (sCHR of COVID-19 pandemic and 2009 influenza pandemic) and moved them to the Supplementary Information 5, Figures S2.

4. The authors seek to improve openness by providing additional data and code, but in multiple places it seems that placeholders still exist in the manuscript, where it reads “available on GitHub at XX”. Please fill in the XX portion.

Response: We have uploaded: 1) the code used in the main analysis and sensitivity analysis; 2) parameters for the code; 3) healthcare seeking behaviors data to GitHub (<https://github.com/zychenfd/Wuhan-disease-burden>). In the revised manuscript, we filled in the XX portion.

Page 34, line 616-617: ...health seeking behavior survey data are available on GitHub at <https://github.com/zychenfd/Wuhan-disease-burden>⁵².

Page 34, line 620-621: ...The R code to replicate the analyses is available on GitHub at <https://github.com/zychenfd/Wuhan-disease-burden>⁵².

5. Please perform a close read and double check for grammatical accuracy and consistency. For example, in the abstract, ‘multiple data source’ should be plural, and in the manuscript, there is some inconsistency in use of past and present tense.

Response: Thanks for the comment. We carefully went through the manuscript and corrected for grammatical and tense errors.

6. As a minor suggestion, in working to address reviewer concerns, the manuscript has continued to grow in length. There may be opportunities to shorten or streamline parts of the paper, particularly the Introduction and Discussion, without sacrificing information and meaning.

Response: Thanks for this comment. To address reviewers’ concerns, the manuscript did grow in length. We slightly shorten the manuscript, with 3696 words in the main text (including Introduction, Results and Discussion), which is less than the word limit of Nature Communications (up to 6,000 words).